# Insights into Asymmetric Liposomes as a Potential Intervention for Drug Delivery Including Pulmonary Nanotherapeutics

**DOI:** 10.3390/pharmaceutics15010294

**Published:** 2023-01-15

**Authors:** Yaqeen Nadheer Al Badri, Cheng Shu Chaw, Amal Ali Elkordy

**Affiliations:** School of Pharmacy and Pharmaceutical Sciences, Faculty of Health Sciences and Wellbeing, University of Sunderland, Sunderland SR1 3SD, UK

**Keywords:** liposomes, asymmetric liposomes, pulmonary, biological membranes

## Abstract

Liposome-based drug delivery systems are nanosized spherical lipid bilayer carriers that can encapsulate a broad range of small drug molecules (hydrophilic and hydrophobic drugs) and large drug molecules (peptides, proteins, and nucleic acids). They have unique characteristics, such as a self-assembling bilayer vesicular structure. There are several FDA-approved liposomal-based medicines for treatment of cancer, bacterial, and viral infections. Most of the FDA-approved liposomal-based therapies are in the form of conventional “symmetric” liposomes and they are administered mainly by injection. Arikace^®^ is the first and only FDA-approved liposomal-based inhalable therapy (amikacin liposome inhalation suspension) to treat only adults with difficult-to-treat *Mycobacterium avium* complex (MAC) lung disease as a combinational antibacterial treatment. To date, no “asymmetric liposomes” are yet to be approved, although asymmetric liposomes have many advantages due to the asymmetric distribution of lipids through the liposome’s membrane (which is similar to the biological membranes). There are many challenges for the formulation and stability of asymmetric liposomes. This review will focus on asymmetric liposomes in contrast to conventional liposomes as a potential clinical intervention drug delivery system as well as the formulation techniques available for symmetric and asymmetric liposomes. The review aims to renew the research in liposomal nanovesicle delivery systems with particular emphasis on asymmetric liposomes as future potential carriers for enhancing drug delivery including pulmonary nanotherapeutics.

## 1. Brief Introduction to Liposomes

The direct delivery of drugs can lead to off-target side effects, poor distribution, and short circulation time due to the breakdown and clearance of the drug [1]. Liposomes are a type of nanocarrier that are described as spherical, microscopic, bilayered vesicles. They have the ability to entrap materials due to the spontaneous assembly of phospholipid molecules when in contact with aqueous media, resulting in the formation of an aqueous inner core surrounded by a closed lipid-based bilayer. Their structure is very similar to the human cell membrane, which is a bilayer mainly consisting of phospholipids; the phospholipids consist of a hydrophilic head and two hydrophobic "tails" derived from fatty acids of different chain length and degree of saturation [2].

Liposomes provide many advantages to the delivery of materials. Liposomes’ ability to entrap several drugs of different properties, having high entrapment efficacy (EE) to reduce dose and provide targeted drug delivery, are some of the main advantages [3]. However, a significant issue with liposomes is the removal from the blood circulation as well as removal by the liver; this is mainly associated with size and charge of the liposomes. Therefore, reduction of size can lead to a longer circulation time [4]. 

Liposomes are proven carriers for pulmonary drug delivery. Based on the literature, pulmonary devices can deliver liposomes (in suspension forms or dry forms) encapsulating drugs into the lung. Pressurized metered inhalers (pMDIs), dry powder inhalers (DPIs), and soft mist inhalers (SMIs) can deliver small amounts of medicine to the lung, whilst medical nebulizers can deliver large liposomal suspension quantities [5]. Arikace^®^ (nebulisable liposomal amikacin suspension) is the first and only FDA-approved liposomal-based inhalable therapy to treat *Mycobacterium avium* complex (MAC) lung disease. Section 2 and Section 3 will provide a brief on conventional liposomal formulations and techniques.

## 2. Liposomal Formulation Composition

Liposomes are generally formulated using non-toxic phospholipids and cholesterol [6]. The phospholipids used when formulating liposomes can be phosphatidylglycerols, phosphatidylcholines, phosphatidylserines, or phosphatidylethanolamines [7]; The choice of the lipid can significantly affect the liposomal properties such as fluidity and charge of the bilayer. Generally, unsaturated phospholipids tend to increase permeability while saturated phospholipids with long acyl chains leads to rigidity and impermeability [6]. The unique structure of liposomes allows them to encapsulate hydrophobic molecules within the phospholipid bilayer and hydrophilic molecules withing the aqueous core (Figure 1). Moreover, liposomal drug delivery can be further enhanced by adding a targeting ligand to liposomes to recognize and bind specific receptors on cells, or by adding biocompatible inert polymers to them such as PEG to reduce phagocyte recognition and increase the liposomal blood circulation half-life [7].

## 3. Conventional Liposomal Formulation Methods

There are various methods used in the preparation of lipid-based nanocarriers such as liposomes [2]. The method of preparation affects critical parameters such as size of vesicle and size distribution, permeability, lamellarity, and entrapment efficiency [9]. Entrapment of compounds is performed by two main techniques; passive loading where drug entrapment occurs during the liposome formation, and active loading where drug entrapment is after the liposome formation. The three classic methods for liposome production are mechanical dispersion, solvent dispersion, and detergent removal. These methods, their advantages and limitations have been comprehensively reviewed [6,9]. The following parts provide a summary of these conventional techniques. 

### 3.1. Thin Film Hydration

Thin film hydration (Figure 2) is one of the oldest and commonly used methods for formulating liposomes in small batch sizes [9]. The main steps for this technique include dissolving the lipids in organic solvent (such as chloroform and ethanol) in a flask, followed by the evaporation of the organic solvent, under vacuum or by using nitrogen stream, to form a dry film of lipids on the inner wall of the flask. The thin film will then be hydrated using a suitable aqueous media while heating the lipids above the phase transition temperature (Tm) and agitating/stirring the formulation. As a result of heating and agitation, the lipid film will get hydrated, swell, and detach from the inner flask wall to form multilamellar vesicles (MLVs). These vesicles tend to be highly heterogenous in lamellarity and size [9]. Moreover, it is difficult to completely remove all toxic organic solvent.

The MLVs can be further processed to control and reduce their size by using downsizing methods such as extrusion, sonication, or high-pressure homogenization [9]. French pressure cell is a method of extrusion that involves applying high pressure and passing the material through a small orifice that transforms MLV into SUVs. It is considered a gentler size reduction technique and only allows for small volume processing [10]. Membrane extrusion is a method that uses a polycarbonate membrane with a defined pore size. Liposomes are passed through this membrane which results in the reduction of the liposome size. Product losses and difficulty to scale up are the main drawbacks. Ultrasonication can be performed by using a bath sonicator or a probe sonicator. This method provides a homogenous suspension of liposomes as well as reducing the size of liposomes by ultrasonic irradiation. However, sonication generates heat, and metal (titanium) particles may be leached off the probe tip to contaminate products and degrade sensitive actives and lipids [11]. Although it reduces size of MLVs, SUVs generated tend to have wide size distribution with lower entrapment efficiency. 

### 3.2. Ethanol and Ether Injections

This method involves dissolving the phospholipid in ethanol (Figure 3), and an aqueous medium is prepared and pre-heated. The ethanol solution containing the dissolved phospholipid is rapidly injected using a needle into the aqueous media containing the material to be entrapped. The mixture requires stirring at high temperature (55–65 °C) to ensure the formation of liposomes. Ethanol will evaporate [12]. Ethanol injection technique is simple and can rapidly form liposomes [13]. This method can form large unilamellar vesicles (LUVs) and small unilamellar vesicles (SUVs) depending on the rate of ethanol injection. Homogenous SUVs are formed when the ethanol volume does not exceed 7.5% of the total formulation volume. Otherwise, heterogenous MLVs are formed. Ethanol is a class 3 solvent which is less harmful, but is volatile and flammable. The presence of residual amount of ethanol in the liposomal dispersion can risk denaturing the entrapped biologically-active macromolecules [9]. 

Ether injection involves dissolving the phospholipid in ether. A solution containing phospholipids dissolved in ether is slowly injected into the aqueous media containing the desired material to be encapsulated. In order to ensure effective evaporation of ether, the mixture is heated to 55–65 °C [12]. SUVs bear similar properties to those fabricated by the ethanol injection. As ether evaporates at a lower temperature than ethanol, it can be efficiently removed in a short time, forming concentrated liposome solutions with relatively good entrapment efficiency [9]. 

### 3.3. Reverse Phase Evaporation

This method involves dissolving the lipids in an organic solvent such as chloroform/methanol (2:1 *v*/*v*) which favours the inverted micelles formation. This is followed by the addition of aqueous buffer to create a water-in-oil microemulsion. Then, the organic solvent is evaporated using a rotary evaporator to form a viscous gel. The gel will then collapse forming liposomes [6]. The presence of large aqueous core of the microemulsions promotes entrapment of especially hydrophilic molecules where the liposomal gels showed a controlled release with a good permeation profile [14]. The technique employs a large amount of organic solvent and solvent extraction process is slow and time consuming [9].

### 3.4. Detergent Removal

This method involves adding a detergent, such as sodium cholate and alkyl glycoside, to phospholipids to solubilise and hydrate the lipids by preventing the hydrophobic portions of the lipids from interacting with the aqueous media forming micelles containing lipid and detergent. Then, the detergent is removed progressively allowing the formation of lipid-rich micelles which spontaneously give rise to unilamellar vesicle formation [9]. The easiest method to remove the detergent is by diluting the suspension using a buffer which also increases the micellar size and polydispersity. However, this technique produces low liposomal concentration and low EE of hydrophobic drugs, mainly due to the dilution step. Alternatively, dialysis technique can be used to remove the detergent. The detergent can also be removed using resin beads, centrifugation, and gel chromatography techniques [9]. 

### 3.5. Microfluidic Devices

A more recent technique involves the use of microfluidic devices for the formation of liposomes; the microfluidic device contains two inlets; the aqueous buffer is added to one inlet and phospholipids dissolved in ethanol is added to the second inlet (Figure 4). The two solutions are mixed through a micromixer, leading to the spontaneous self-assembly of the liposomes due to the change in polarity of the solution [15]. The microfluidics device produces liposomes under ambient process temperature without heating the lipid above its transition temperature as is required in the lipid hydration technique. It also generates a laminar flow pattern for liposome formation in a controlled manner. There are different designs of the micromixers that provide an efficient mixing within short retention time in the mixing chamber, which has been extensively reviewed [16]. Some drawbacks are the needs to remove residual organic solvent and cost of renewing microfluidic cartridges. This method can be made into a ‘lab-on-chip’ system and an adopted continuous flow process for potentially the large-scale liposome production. 

Although the methods described above are commonly-used methods for the effective formation of liposomes, each method has certain challenges (Table 1). The main drawback of these methods is that they are only able to formulate symmetrical vesicles, meaning that the liposomes contain the same lipid composition in the outer and inner leaflets [17]. This is considered a limitation in the formation of liposomal carriers as artificial bilayer carriers are mostly designed for the aim of mimicking biological membranes. However, biological membranes are highly asymmetrical with different lipid compositions in each leaflet. Thus, to improve the mimicking of biological membranes, liposomes need to be formulated with an asymmetrical nature [18]. 

## 4. Nature of Biological Membranes

Biological membranes are typically formed from a phospholipid bilayer which contains hydrophilic heads facing outwards and hydrophobic acyl chains facing each other “inwards” [19]. The plasma membrane contains various types of phospholipids with different properties including melting points, headgroups, intrinsic curvature, saturated/unsaturated acyl chains, and cholesterol. As shown in Figure 5, these phospholipids are distributed asymmetrically throughout the plasma membrane [20]. The lipid layer whose headgroups are facing the outer environment form the outer leaflet (exofacial layer) and the lipid layer whose headgroups are facing the inner components of the membrane form the inner leaflet (cytofacial layer) [21]. 

In 1972, Mark Bretscher [23] published the first report that talked about partial lipid asymmetry in biological membranes. He has found that the outer and the inner leaflets are composed of different lipids. Thus, bio-membranes have asymmetrical features [23]. Eukaryotic cell membranes are bilayers of asymmetric lipids where the outer layer consists of sphingomyelin (SM) and phosphatidylcholine (PC) while phosphatidylinositol (PI), phosphatidylethanolamine (PE), and the negatively-charged phosphatidylserine (PS) are found in the inner layer [24]. Additionally, cholesterol is a key lipid that is present in the phospholipid bilayer; it forms around 40 mol% of lipids in the synaptic plasma membrane and is distributed in both leaflets in an asymmetric fashion [21]. Bacterial membranes were also shown to have an asymmetrical features, but with different compositions; the inner leaflet predominantly consisting of PI and PE while outer leaflet mainly consisting of phosphatidylglycerol (PG) [25,26]. Lipid asymmetry is maintained by specific proteins, these proteins have a specific structure that can help in the movement of lipid molecules across the leaflets. Flip/flopases are mainly responsible for moving lipids across the leaflets, while scramblases use an energy independent and a non-selective mechanism to mediate the trans-bilayer movement [27]. Moreover, other forms of asymmetry are present within the membranes as follows:

### 4.1. Geometric Asymmetry

A common source of asymmetry occurs due to vesicle size. As the vesicle diameter decreases, an increase in the difference between the leaflets’ surface areas is noticed due to unequal number of lipid molecules that exist in bilayer leaflets [28]. Moreover, lipid intrinsic curvature leads to lateral and transverse lipid separation [29]. According to the shape parameter (S) equation:$$S=\frac{V}{l_{c}a_{o}}$$
where *V* is the volume per molecule, *l_c_* is the length of the fully-extended acyl chain, and *a_o_* is the optimum area per molecule at the lipid/water interface. If S > 1, then an inverted cone shape will form which prefers negative curvature. While if S < 1, a cone-like shape will form with a preference to positive curvature. If S = 1, then a cylindrical shape will be adopted which corresponds to a neutral curvature [28]. SM and PC were found to have regions of positive or neutral curvature, while PE and PS tend to form regions of negative to neutral curvature. Thus, this could be the explanation to the presence of PS and PE predominantly in the inner monolayer of the plasma membrane [24,30].

### 4.2. Cholesterol Distribution

The distribution of cholesterol within the membrane leaflets is still debated, however, some studies suggest that cholesterol molecule preferentially resides in the inner leaflet and has an asymmetric distribution within the plasma membrane [31]. This speculation was based on a study by Wang et al. [32] which demonstrated that cholesterol has an affinity for areas with high curvature. It is suggested that PE, which mainly resides in the inner leaflet, has regions of high negative curvature; this could be the reason that cholesterol is preferentially drawn to the inner layer [28]. 

### 4.3. Charge

As discussed previously, the phospholipid distribution within the cell plasma membrane has an asymmetric nature. Due to this asymmetry, the charge in the outer and inner leaflets differs. Neutral lipids such as SM and zwitterionic PC are mainly located within the outer layer of the plasma membrane. While anionic phospholipids, e.g., PS, PE, and PI, tend to be present within the inner layer [33].

### 4.4. Exosomes

They are small, extracellular vesicles that are released from cells [34] as means of communication with other cells [35,36]. Exosomes are unique vesicles which were found to contain an asymmetrical lipid membrane with similar membrane structure to eukaryotic cells [37]. 

The asymmetry of lipid membranes has an effect on various membrane properties, such as stability, shape, surface charge, membrane potential, and permeability [28]. The loss of this asymmetry has been associated with consequences. PS levels have a significant effect on cells, for example, during apoptosis, PS move to the outer leaflet, exposing themselves to macrophages, which signals for macrophages to degrade the cell [38]. Moreover, PS acts as an important co-factor for various enzymes within the membrane, for example, protein kinase C [39] and the externalisation of PS leads to promotion of the coagulation cascade [40]. 

The cell membrane was found to have the ability to form platforms (rafts) which can help in cell signalling and trafficking [41]. The cell membrane consists of two immiscible phases, ordered phase “Lo”, which is rich in cholesterol that is tightly bound to high-melting lipids such as SM [42,43] and disordered “Ld” phase [44] which is rich in unsaturated acyl chains [18]. The lipids in the lipid disordered state are less tightly packed when compared to lipids in the ordered/gel state [45]. To form platforms, the membrane segregates the constituents with the help of the two-phase immiscibility, and form compartments (rafts) which are rich in sphingolipids, cholesterol, and proteins [41]. 

Ordered domains formed in the outer leaflets can be isolated from cell lysates and model membranes and can be detected using fluorescence quenching [46]. While the formation of ordered domains within the inner leaflet, using phospholipids that are predominant within the inner leaflet, might not be possible [47], there are speculations on the presence of ordered domains within the inner leaflet [48]. Recent studies, using asymmetric model membranes, have revealed that ordered domains formation in one leaflet can lead to the formation of ordered domains in the other leaflet [49]. Moreover, the tuning of lipid mixtures can induce or suppress domain formation across leaflets, suggesting interleaflet interactions [50]. Interestingly, interleaflet communication can be further suggested as the formation of membrane domain in the outer leaflet can influence the inner leaflet-associated proteins organisation during the process of signal transduction [28]. Additionally, the inner leaflet components can sense the outer leaflet components and respond to their physical state. However, this coupling is still not fully understood, and more studies are needed to understand this phenomenon [28]. This will lead to the next sections on asymmetric liposomes as drug delivery systems, their advantages, formulation methods, formulation challenges, and their potential applications. 

## 5. Advantages of Asymmetrical Liposomes

Greco et al. [51] via Anderson et al. [52], formulated asymmetric liposomes that were like apoptotic bodies that kill *Mycobacterium tuberculosis* bacteria free from antibiotic drugs, aiming to alleviate the incidence of antibiotic resistance in tuberculosis treatment. The asymmetric liposomes were formed from L-α-phosphatidylserine (PS) distributed at the outer membrane to mimic apoptotic bodies (hence promote uptake by macrophages) and L-α-phosphatidic acid (PA) distributed at the inner layer to improve vesicle trafficking and fusion within macrophages while reducing the inflammatory response. Favoured uptake into macrophages more than alveolar epithelial cells was noted. The asymmetric, apoptotic body-like, liposomes were effectively internalised by macrophages and led to the induction of Ca^2+^, which was related to the inhibition of both bacterial growth and inflammatory responses. The asymmetric liposomes were administered intranasally to *Mycobacterium tuberculosis* infected BALB/c mice—more information can be found in Greco et al. [51]. This study shows the promise in application of a special type of liposome, “asymmetric liposomes”, for treating pulmonary infection diseases. Hence, this section and the following Section 6, Section 7, Section 8 and Section 9 will give details on asymmetric liposomes. 

Asymmetric liposomes are liposomes that contain different lipid composition in the outer and inner leaflets [53]. Therefore, they allow for the possibility of enhancing the properties of the inner and outer leaflets independently; lipids that can maximise entrapment efficiency and reduce leakage can be used in the inner leaflet and different lipids can be used in the outer leaflet to enhance drug delivery and liposomal stability [3]. In a study done by Whittenton et al. [53], inverse emulsion technique was used to formulate asymmetric liposomes that contain cationic lipids DMPC (1,2-Dimyristoyl-sn-glycero-3-phosphocholine) and DOTAP (Dioleoyl-3-trimethylammonium propane) in the inner leaflet and neutral/negatively-charged lipids DMPC/POPC (1-palmitoyl-2-oleoyl-sn-glycero-3-phosphocholine) with NBD-PC (1-oleoyl-2-[6-[(7-nitro-2-1,3-benzoxadiazol-4-yl)amino]hexanoyl]-sn-glycero-3-phosphocholine)) in the outer leaflet. These liposomes were able to entrap negatively-charged polynucleotides. Moreover, the asymmetric liposomes structure can be adjusted based on the molecule used; a study done by Li and London [3] entrapped Doxorubicin, a cationic drug using different combinations of cationic lipids DOTAP, POePC (1-palmitoyl-2-oleoyl-sn-glycero-3-ethylphosphocholine)) and anionic lipids POPG (1-palmitoyl-2-oleoyl-sn-glycero-3-phospho(1′-rac-glycerol)), POPS (1-palmitoyl-2-oleoyl-snglycero-3-phospho-L-serine), and POPA (1-palmitoyl-2oleoyl-sn-glycero-3-phosphate)) in the outer and inner leaflets in different combinations. The study has shown that asymmetric liposomes containing anionic lipids in the inner leaflet, regardless of the lipid present in the outer leaflet, had the highest entrapment of doxorubicin as well as slowest leakage. 

Table 2 compares between symmetric and asymmetric liposomes regarding their compositions, production methods, and physicochemical characteristics.

## 6. Considerations Related to Formulating Asymmetrical Liposomes

There are parameters which should be taken into consideration when formulating asymmetric liposomes:

### 6.1. Maintenance of Asymmetry

A main challenge to formulating asymmetric liposomes is the flip/flop of the lipids and loss of asymmetry. It was suggested that the rate of flip/flop is affected by the thickness of the bilayer, where a reduced flip/flop rate was seen in bilayers with a thicker hydrocarbon region with phospholipids containing longer acyl chains. The rationale behind this link can be due to the high energy requirement to move a polar headgroup through a longer hydrophobic path of the thick membrane [28].

### 6.2. Interleaflet Coupling

Cheng and London [48] have studied the effect of temperature and curvature on interleaflet coupling of asymmetric large unilamellar vesicles (LUV). LUVs have reduced membrane curvature as compared to small unilamellar vesicles (SUVs) and it is a closer mimic to the plasma membranes. The study found that the properties of LUVs were similar to that of SUVs, thus, suggesting that curvature does not significantly affect interleaflet coupling. However, the interleaflet coupling was significantly affected by temperature. At ambient temperature, strong interleaflet coupling was observed as SM was exchanged into the outer leaflet and produced asymmetry. However, as temperature approaches 37 °C, interleaflet coupling became very weak. Additionally, it was observed that asymmetric LUVs showed a higher order compared to symmetric LUVs using the same lipid composition, which could also indicate interleaflet coupling [48]. A considerable increase in inner leaflet order was seen due to the presence of a highly-ordered outer leaflet [48]. In asymmetric vesicles containing SM on the outer leaflet, SM was able to form an ordered state; the thermal stability was significantly higher than symmetric liposomes with similar lipid composition [48]. 

### 6.3. Hydrophobic Acyl Chains

The stability of the asymmetry was found to be related to the acyl chain structure; the structure of the acyl chains influenced the transverse diffusion (flip-flop). Moreover, it was deduced that the headgroup structure of the phospholipids can influence whether the asymmetry is full or partial [54]. Maintenance of asymmetry was significantly prevented when overly short or two polyunsaturated acyl chains were present; this could explain why phospholipids with these properties are not abundant in biological membranes [55]. Short acyl chains are unable to form a sufficiently stable bilayer. Moreover, two polyunsaturated acyl chains are prone to oxidation and are extremely sensitive.

### 6.4. Charge

Lipids with only one charge, e.g., anionic, can cross the membrane more readily in an uncharged state which can occur due to protonation or complexation with Na^+^ or K^+^. This gives rise to partial asymmetry. Additionally, the free energy is raised due to the repulsion between the negative charge of the neighbouring anionic lipids which exacerbated the tendency to flip between the leaflets. The presence of Phosphatidylethanolamine (PE), a zwitterionic phospholipid, can lead to more stable asymmetry. This can be due to the presence of multiple charges that can lessen this repulsion and reduce the tendency to flip. Moreover, PE has smaller headgroup size which can help in reducing steric clashing with phospholipids with larger headgroups such as PC [55]. 

### 6.5. Cholesterol Level

Cholesterol plays a significant role in modulating membrane permeability. It was found that liposomes containing 100% POPC had significantly higher permeability than those containing 60% POPC and 40% cholesterol [56]. Moreover, the formation of ordered domains is more stable in vesicles containing 25 mol% cholesterol than those without cholesterol. This shows the importance of cholesterol in forming and stabilizing ordered domains [18]. To control the cholesterol levels within the asymmetric vesicles, cyclodextrin-exchange method can be performed using (2-hydroxylpropyl)-α-cyclodextrin [HPαCD]. HPαCD has a small ring size with little to no affinity to cholesterol and an affinity for phospholipid. This allows for cholesterol to be embedded in the acceptor vesicle, as will be explained later, before the lipid exchange process. However, the use of HPαCD can have some complications such as less affinity for certain phospholipids compared to others [57].

## 7. Current Formulation Techniques for Asymmetrical Liposomes

The different types of methods used to formulate asymmetric liposomes can be divided into two main categories based on the size of the formed vesicles—nano-sized and cell-sized. 

### 7.1. Nano-Sized Asymmetric Liposomes Formulation Techniques

#### 7.1.1. Cyclodextrin Exchange Method

Cyclodextrins (CD) are defined as cyclic oligosaccharides with two distinct regions containing a hydrophilic exterior and a hydrophobic interior [58]. Cyclodextrin exchange (Figure 6) is a novel method, developed by Prof. London and co-workers, that leads to the formation of asymmetric liposomes with different lipids/charges in the inner and outer leaflets of the liposomes. The asymmetric liposomes can be prepared as vesicles containing lipids of different charges, e.g., zwitterionic, cationic, or anionic [3]. During the preparation method, two different vesicles must be formulated, a donor and acceptor vesicles. The donor vesicle is formulated as a multi-lamellar vesicle (MLV) and added to cyclodextrin; the acceptor is formulated as a unilamellar vesicle with sucrose entrapped; sucrose is used to aid isolation during the centrifugation. This is achieved by preloading the acceptor vesicle with a high concentration of sucrose, e.g., 25% *w*/*w* which creates a significant density difference between the donor MLVs and the acceptor unilamellar vesicles when the vesicles are suspended in phosphate-buffered saline (PBS) [56].

The two vesicles are then added together, and with the help of CD, lipid exchange process occurs leading to the formation of asymmetric liposomes. The CD has only the ability to exchange the outer layer of the acceptor vesicle, therefore, the inner layer of the acceptor vesicle is unaffected by the exchange process. Sucrose will help in the separation of asymmetric vesicle during the ultra-centrifugation process; the supernatant will contain all the impurities while the asymmetric liposomes will be pelleted at the bottom of the tube. The extracted vesicles are then resuspended with buffer. Once the asymmetric liposome is formulated, the outer leaflet will contain the phospholipid composition of the donor vesicle and the inner leaflet will contain that of the acceptor vesicle [3]. Moreover, a concentration of 40 mol% of cholesterol was found to be the ideal concentration to get the highest yield of asymmetric liposomes obtained after centrifugation [3] which is similar to the amount of cholesterol found in synaptic plasma membrane [21]. 

The exchange between the donor and the acceptor vesicles must be at a 1:1 ratio i.e., for every one lipid removed from the acceptor, one lipid is added to the acceptor, otherwise, a stress will be induced between the leaflets which can lead to vesicle rupture [3]. The asymmetric vesicles were able to remain stable for 48 h [3]. This issue, from our point of view, can be solved by lyophilisation technique, however, stability and efficacy investigation of asymmetric liposomes after drying via lyophilisation is needed. 

Although this technique provides promising results, adding a high concentration of sucrose to the acceptor vesicle can be associated with osmolarity gradient related membrane tension and potential structural perturbations due to lipid–sucrose interactions [60]. To overcome this issue, Heberle et al. [60] modified the cyclodextrin-exchange method by loading the sucrose into the donor MLVs instead, this allowed for the removal of sedimented sucrose-loaded MLVs after exchange using low speed centrifugation; then the removal of the remaining cyclodextrin molecules using a centrifugal concentrator. 

Further modifications were carried out by Markones et al. [61] to improve the degree of donor lipid incorporation into the final asymmetric vesicle. Instead of using donor MLVs, a donor lipid–cyclodextrin complex was used during the exchange process. ζ-potential measurement was undergone to determine the extent and stability of the asymmetry which resulted in an asymmetry stable for 14 days at 20 °C. Additionally, Markones et al. [62] was able to formulate asymmetric proteoliposomes using a five-step method which involves formulating a unilamellar vesicle with the desired lipids then adding the desired proteins (Na^+^/H^+^ antiporter NhaA transmembrane protein was used in this study), adding a donor lipid–cyclodextrin complex to initiate the exchange process, followed by the formation of asymmetric proteoliposomes. Finally, validation of the asymmetric proteoliposomes is achieved by measuring the ζ potential, and the protein is characterized by performing a fluorescence-based protein activity assay.

In addition to proteins incorporation into asymmetric liposomes, the addition of peptides was also tested. Doktorova et al. [63] studied the peptide-membrane interactions by looking at the effect of peptides on membrane asymmetry by using asymmetric LUVs. Gramicidin, a peptide, was used to test this effect; the results have indicated that the rate of flip-flop was increased by a factor of 3. This was further confirmed by Nguyen et al. [64] who studied the effect of gramicidin and other peptides (alamethicin, melittin, or the pH low insertion peptide (pHLIP)), and shown that the flip-flop rate with gramicidin was increased while the asymmetry was immediately destroyed when the other peptides were added to the asymmetric LUV. 

#### 7.1.2. Reverse Phase Evaporation

In a study conducted by Mokhtarieh et al. [65], using siRNA, asymmetric liposomes were formulated using a modified reverse phase evaporation method; where two inverted micelles with different phospholipid compositions were prepared separately then mixed; the inner micelle contained 1,2-dioleoyl-3-dimethylammonium-propane (DODAP) and 1,2-dioleoyl-sn-glycero-3-phosphoethanolamine (DOPE) and were dissolved in ether and citrate buffer, while the outer micelle contained 1,2-distearoyl-sn-glycero-3-phosphocholine (DSPC), DOPE, polyethylene glycol1,2-distearoyl-sn-glycero-3-phosphatidylethanolamine (PEG-PE), and cholesterol and were dissolved in ether and HEPES-buffered saline (HBS)/ethanol. Following mixing, ether evaporation and dialysis were performed to formulate the asymmetric liposomes. These liposomes were then pegylated and antibodies/peptides were conjugated. Analysis techniques involved serum stability and toxicity as well as nuclease protection assay. However, no specific analysis technique was conducted to confirm asymmetry. The result from this study showed that this method achieved more than 90% encapsulation efficiency and an average size of 200 nm, although no comparison to symmetric liposomes was done. Moreover, the liposomes were able to protect the siRNA from RNAases for up to 24 h. PEGylation lead to the prevention of aggregation as no aggregation was found in the siRNA/Asymmetric liposomes and serum mixture. 

Extrusion has the ability to effectively minimize vesicle size, however, this reduces liposomes encapsulation [4]. Mokhtarieh et al. [4] modified this method by adding ethanol to reduce the vesicle size. Ethanol was added immediately after the liposome formation and before the complete removal of ether. The results have shown that ethanol treatment has the ability to reduce the vesicle size to 100–200 nm without affecting the liposome’s structure and properties. Moreover, the encapsulation efficacy of the liposome was not affected by the size reduction. 

#### 7.1.3. Ca^2+^ Induced Asymmetry

This method involves the use of Ca^2+^ ions to cause asymmetry. Sun et al. [66] formulated LUVs containing DPPC (1,2-dipalmitoyl-sn-glycero-3-phosphocholine) and DOPS (1,2-dioleoyl-sn-glycero-3-phospho-l-serine) then added 0.5mM of Ca^2+^ and incubated the mixture at 70 °C for about 40 h. This has driven the Ca^2+^ ions to bond to two PS molecules headgroups and forming a PS-PS-Ca^2+^ complex which favours negative curvature and leads to the migration of PS to the inner layer [66]. Fluorescence quenching technique was used to confirm asymmetry and the asymmetry was stable for several days at room temperature. The parameters of this method were further explored by Guo et al. [67] who looked at the effect of temperature, lipid content, and vesicle size. It was shown that increasing mol% of PS lead to the decrease of asymmetry; asymmetry was not affected by temperature when reducing temperature from 70 °C to 50 °C; increasing the size of the vesicles lead to a reduction in asymmetry. Although this is a promising method, only negatively-charged lipids can be used to form a complex with Ca^2+^, and furthermore, the method requires a very long incubation time (~40 h). 

#### 7.1.4. The Use of Enzymes

The asymmetry of Phosphatidylserine (PS) has the most pronounced effect on the cell, thus, keeping the levels of PS stable within the membrane is crucial. The level of PS in the outer leaflet can be between 0–3.2 mol%, while in the inner leaflet it can be as high as 20 mol% [68]. A study by Drechsler et al. [68] has used a unique method to formulate asymmetric liposomes with PS content similar to that of biological membranes i.e., low PS level in the outer leaflet and a high PS level in the inner leaflet. Symmetric liposomes were formulated, then Phosphatidylserine decarboxylase (PSD) enzyme was used to decarboxylate the outer leaflet’s anionic PS to neutral PE; since PSD is water soluble, it cannot penetrate the liposomal membrane. Thus, it only converts the PS on the outer leaflet while PS in the inner leaflet remains unchanged. The change in the outer leaflet was analysed by measuring ζ-potential which was dropped from −50 mV to −23 mV. Moreover, high-performance thin layer chromatography, HPTLC, was used to confirm asymmetry by measuring the content of PS after PSD treatment. The asymmetry remained unchanged for 4 days at 20 °C. Although this technique mimics the PS level in biological membranes, it converts PS to PE in the outer leaflet while PE is predominantly found in the inner layer of biological membranes

Phospholipase D was used by Takaoka et al. [69] to convert PC to PS and PE and formulate asymmetric liposomes using the same approach. Although this is an effective technique to formulate asymmetric liposomes, only lipids that interact with the enzymes can be used. 

### 7.2. Cell-Sized Asymmetric Liposomes Formulation Techniques

#### 7.2.1. Inverted Emulsion Technique

This method was proposed by Pautot et al. [70] to formulate asymmetric liposomes (Figure 7). The technique involves the assembling of each leaflet independently. First, a water-in-oil emulsion (w/o) is formed, and the lipid used to form inner leaflet of the asymmetric liposomes is used to stabilise this emulsion. The emulsion phase is then layered over an intermediate phase of the same oil but containing the outer leaflet’s lipids. The intermediate phase is heavier than the emulsion phase and thus the emulsion phase will be on top. The intermediate phase will be placed over an aqueous phase. The outer leaflet’s lipids found in the intermediate phase will form a monolayer between the intermediate phase and the aqueous phase. The water droplets in the w/o emulsion, that are covered with the inner leaflet’s lipids, are heavier than the oil in the emulsion and intermediate phase. Thus, water droplets will sink towards the aqueous phase and pull the lipid monolayer present between the intermediate and aqueous phases to form asymmetric vesicles in the aqueous phase. Centrifugation is used to accelerate the sinking process. This method was further developed by utilizing microfluidics [71]. 

Although this method is considered less violent in terms of vesicle construction than the cyclodextrin-exchange method, the final vesicle will potentially contain organic solvent which is the main drawback of this method. According to the study done by Whittenton et al. [53], inverted emulsion technique can successfully form asymmetric liposomes. However, the inverse emulsion to liposome conversion had a low yield with this technique, especially when dodecane and mineral oil are used. The use of squalene had higher yield which could be due its higher viscosity and reduced interfacial tension [53]. The asymmetry was confirmed using fluorescent NBD-labelled lipids as the fluorescent label was in the outer leaflet. For the inverted emulsion in a centrifugation field technique to be successful, the oil–water interface must be fully equilibrated [72].

#### 7.2.2. Microfluidics

Microfluidic devices can be a useful tool for the formation of asymmetric liposomes; Hu, et al. [73] have proposed a two-step route which combines microfluidic generation with emulsion transfer to form asymmetric giant unilamellar liposomes. The formation involved two independent steps. Firstly, the microfluidic device was used to form the first monolayer; the device contained a multiphase droplet flow centrifugation which consists of a continuous oil stream which allows for the formation of water droplets; then a vessel containing a layer of oil over a layer of water is used to dispense the water droplets into. The second step involves transferring the droplets through a second oil–water interface by centrifugation. This will lead to the formation of the second monolayer, similar to the inverted emulsion technique mentioned above. Different oil phases are used to dissolve different lipids which allows for the control of the resulting lipid bilayer. Fluorescence quenching, biotin-binding assay, and annexin V assay were used to confirm asymmetry. Lu et al. [74] engineered asymmetric vesicles using combinations of novel microfluidic techniques. The method involved four main steps: (1) highly uniformed w/o emulsion formed and stabilized by the “inner-leaflet” lipid, (2) the “inner-leaflet” lipid is replaced by the “outer-leaflet” lipid surrounding the w/o emulsion, (3) this creates a w/o/w double emulsion template which encapsulates the w/o emulsion, and (4) the “outer-leaflet” lipid solution is removed from the intermediate layer of the double emulsion. The results from this study have shown that the membrane asymmetry was maintained for over 30 h; 80% of the asymmetric vesicles remained stable for at least 6 weeks, additionally this method was able to improve the size variation control. 

More recently, Ghazal et al. [75] has combined a microfluidic platform based on hydrodynamic focusing on the thiol-ene chip with synchrotron small-angle X-ray scattering (SAXS) to examine the continuous production of multilamellar vesicles (MLVs) of nano-dimension. Due to an uneven distribution of the two embedded lipid molecules at the lipid interfacial-water area, formation of an asymmetric bilayer could not be ruled out and it has been suggested that the growth of asymmetric feature is a time-dependent process. Pulsed jet flow also utilising the microfluidic template has recently showed to produce a nano-dimension asymmetric liposome [37]. There is an abundance of evidence supporting the production of giant-sized vesicles by microfluidic devices while production of nano-size equivalence is feasible. Further advancement in microfluidic devices is required in order to control the size of produced MLVs and bilayer asymmetry properties. 

#### 7.2.3. Hemifusion

Enoki and Feigenson [76] have formulated asymmetric giant unilamellar vesicles (GUV) using hemifusion (Figure 8). This technique involves formulating a giant unilamellar vesicle (GUV) by electroformation and applying a red fluorophore for visualization by confocal microscopy. A supported lipid bilayer (SLB) is formulated separately, and a green fluorophore is added. Then the GUV is put in contact with the SLB, where a fusogenic agent (Ca^2+^), as calcium chloride, induces hemifusion; this will lead to lipid exchange between the SLB and the outer leaflet of the GUV. The GUV is then detached from the SLB by adding EDTA to chelate the calcium and stop hemifusion. A new asymmetrical GUV is formed which contains the GUV lipids in the inner leaflet and the SLB lipids in the outer leaflet. The asymmetry is confirmed by measuring the intensity of each fluorophore in the asymmetric GUV in contrast to the symmetric GUV. This method was able to approach 100% asymmetry and preserve vesicle content [76]. Moreover, the line tension of domains was investigated and showed that asymmetric liposomes with DOPC-rich outer leaflet has lower line tension when compared to their symmetric counterparts [20]. 

#### 7.2.4. Pulsed-Jet Flow

This method involves the formation of a lipid tube which is then deformed and leads to the formation of asymmetric vesicles. Kamiya et al. [77] have used this method to formulate cell-sized giant vesicles (GVs). Microfluidic flow is applied via a jet nozzle to formulate a micro-sized lipid tube; the lipid tube contained one type of lipid in the outer layer, n-decane organic solvent in between, and a different lipid in the inner layer (DOPS and DOPC were used). Then, the tube was deformed by applying sinusoidal undulation, this in turn forms two types of asymmetric vesicles—one with a diameter of ~100–200 µm and vesicles with ~3–20 µm diameter. Confocal Raman scattering microscopy was used to study these vesicles and was shown that the latter contains only a small amount of organic solvent within the monolayers of the membrane. The 3–20 µm-sized vesicles were then used to study lipid–lipid and lipid–protein interactions. The results have shown that asymmetric vesicles containing DOPC in the inner layer and DOPS/DOPC in the outer layer has led to the increase of membrane reconstitution ratio of the proteins into lipid membranes. This method was able to overcome the remaining of residual organic solvent issue associated with other techniques such as inverse emulsion technique. The presence of organic solvent can affect the long-term stability of vesicles and leads to vesicle rupture within few days. However, this method was able to keep vesicles stable for at least 7 days [77].

Intracellular vesicles within the cells play an important role in homeostasis and regulation of metabolism and they were found to have an asymmetric membrane that is different from the plasma membrane. Developing this system can create an in vitro model to improve the understanding of this synaptic system [78]. Kamiya et al. [78] modified the pulsed-jet flow method to form a vesicle-in-a-vesicle system by using a triple-well device which mounts two separators. To generate the inner vesicles, the separator between first and second wells has an opening sized 100 µm; to generate the cell-sized vesicles, the separator between the second and third wells has an opening sized 500 µm; the inner vesicle would be inserted into the cell-sized vesicles. To confirm asymmetry, fluorescence quenching method was conducted by using phospholipid-conjugated NBD in the outer leaflet and imaging using confocal microscopy; moreover, fluorescence annexin V binding assay was used to measure the asymmetry of the inner leaflet of the cell-sized vesicle and the outer layer of the inner vesicle [78].

Further modification of this method by Kamiya et al. [37] allowed for the formation of nano-sized asymmetrical lipid vesicles using pulsed-jet flow. This was performed by using asymmetrical planar lipid bilayer with increasing the application of pressure and duration. The lipid bilayer used mimics exosomes; thus, it can provide a useful tool for exosome-like delivery systems which can improve the lipid vesicle interaction with living cells. To confirm asymmetry, the technique involved using streptavidin (biotin)-conjugated gold colloids which can bind to biotin-conjugated phospholipids on the outer leaflet. 

## 8. Challenges Associated with Formulating Asymmetric Liposomes

Although several promising techniques have been discussed for formulating asymmetric liposomes, these methods have their own limitations (Table 3). Additionally, it may not be possible to compare these methods head-to-head as the published literature available does not measure the same parameters such as encapsulation efficiency, stability of asymmetry, degree of asymmetry, and asymmetric vesicle stability. Moreover, an important but overlooked parameter to consider when formulating asymmetric liposomes is differential stress. Differential stress is described as the optimal lipid packing density imbalance between the leaflet leading to residual leaflet and it affects vesicle and asymmetry stability tension [79]. Symmetric vesicles tend to have tensionless (zero tension) leaflets, while asymmetric leaflets tend to be under differential stress, in other words, have a non-zero leaflet tension [59]. 

## 9. General Analytical Techniques

When formulating asymmetric liposomes, it is essential to have confirmatory methods to prove the asymmetry. Measuring zeta potential is one of the methods that has the ability to detect asymmetry of ionic lipids within the liposome by measuring the charge of ionic lipids only in the outer leaflet of symmetric and asymmetric liposomes [61]. 

A novel method developed by London and co-workers involves using a cationic fluorescent probe to bind to the outer layer [3]. DPH (diphenylhexatriene) and TMA-DPH (trimethylammonium diphenylhexatriene) fluorescence measurements are used in the confirmation of asymmetry [18]. DPH can dissolve throughout the bilayer, while TMA-DPH is restricted to the outer leaflet as it does not flip rapidly between inner and outer leaflets. TMA-DPH involves the use of a fluorescence probe that has a positive charge; this probe does not cross the membranes easily and its binding is dependent on the outer-leaflet charge. When the membrane has a negative charge, the probe will have a high level of binding and as it inserts into the hydrophobic core of the liposomal bilayer; the fluorescence will significantly increase which allows for the detection of charge [3].

High-performance thin layer chromatography (HPTLC) can be used to quantify the lipid composition of the liposomes and that can be useful in monitoring the change in membrane compositions in formulating asymmetrical liposomes [80]. The phospholipid composition change after lipid exchange was measured via HPTLC in a study by Li and London [3]. Moreover, HPTLC was used to quantify the change of PS concentration after the application of PS decarboxylase enzyme. 

Fluorescence quenching is another method that can be used to determine asymmetry by measuring the fluorescence intensity of the tagged phospholipids such as NBD-conjugated lipids [70]. The use of fluorescence dye [20] is another effective method of confirming asymmetry. Fluorescently tagged annexin V is a type of fluorescence analysis that can be used to bind to PS then viewed by confocal microscopy to detect asymmetry [73,77].

Recently, neutron and X-ray scattering techniques have shown how the structural and dynamical properties of each leaflet respond to changes in lipid compositions. These analytical techniques are indispensable for the development and characterization of the complex asymmetric vesicles [81]. Analysis techniques used by Heberle et al. [60] involved the use of small-angle neutron scattering (SANS) and nuclear magnetic resonance (NMR) experiments by preparing the liposomes in different buffers containing different concentrations of deuterated water to study the influence of the fluid inner leaflet on the more ordered outer leaflet. SANS method, with subnanometer resolution, is an effective method that has the ability to determine bilayer structure [60]. Furthermore, gas chromatography/mass spectroscopy was used as an exchange efficiency quantification method and the degree of asymmetry was measured using ^1^H NMR and Pr^3+^ shift reagent which shifts resonance when binding to the choline methyl groups in the outer leaflet; this only occurs in the outer layer because Pr^3+^ does not have the ability to cross the membrane [59]. 

Conjugation is another useful technique used to confirm asymmetry. Conjugating certain materials with the phospholipids can help detect asymmetry. Kamiya et al. [37] used streptavidin (biotin)-conjugated gold colloids to confirm asymmetry of the vesicles. Biotin was conjugated with the lipids in the outer leaflet and the streptavidin-gold colloids were added to the solution; TEM was used to visualise the streptavidin-gold colloids (small, black spots) attached to the outer leaflet lipids and confirm asymmetry. Moreover, when the streptavidin-gold colloids were added to the biotin linked to the inner leaflet lipids, no gold colloids appeared in the TEM image, indicating that all the inner leaflet lipids were indeed in the inner leaflet. 

## 10. Potential Benefits to Asymmetrical Liposomes in Genetic Material Delivery

Genetic material delivery into the human body requires a suitable carrier to protect the nucleic acids and allow them to be transported safely to the targeted cells; this is because naked genetic materials are significantly susceptible to degradation. Immune response sensitization, phagocytosis, serum nucleases degradation, and rapid renal clearance, in addition to low cellular uptake and target specificity, are vulnerabilities that make naked genetic material delivery highly unsuitable and ineffective and can be eliminated from the body rapidly [82]. To successfully deliver genetic material to targeted cells, the carrier must form a stable complex with the encapsulated genetic material; the complex must be able to survive in the blood circulation by avoiding early recognition by macrophages and reaching the targeted cells. Opsonins are serum proteins that attach to liposomes and target them for removal by macrophages [83]. Once inside the cell, the liposomal carrier must have the ability to escape the endosomal degradation “endosomal escape”. Moreover, the process of genetic material delivery should pose minimal side effects and enhanced therapeutic action [83].

Nucleic acids used in lipid carriers are generally divided into three main types, small DNA (Oligodeoxynucleotides) or chemically synthesised related molecules, large DNA molecules (Plasmid DNA), and RNAs (small interfering RNA “siRNA”, messenger RNA “mRNA”, and Ribozymes) [84]. Since nucleic acids are negatively-charged, they require a positively-charged carrier to be able to bind to them. These carriers that form a complex with the nucleic material can be formed from positively-charged polymers (polyplexes) [85] or cationic lipids (lipoplexes) [86]. However, the main disadvantages of the use of positively-charged carriers are the removal by the reticuloendothelial system (RES) as well as nonspecific interactions with predominantly negatively charged blood components, thus, leading to accumulation at the primary organs [87]. Therefore, based on the asymmetric liposome advantages discussed, the use of asymmetric liposomes can potentially help in overcoming issues associated with the current genetic material delivery methods

## 11. Conclusions

Nano-based drug delivery systems have become an attractive approach for treating various diseases including respiratory diseases. Liposomal nanotherapeutics have many advantages for specificity and other characteristics over conventional therapies. Inhalable liposomes have proved effective based on the literature and clinical trials which led to the approval and existence of Arikace^®^ in the market (FDA-approved amikacin liposomal suspension based inhalable therapy) and there will be more in the clinic in the future. Due to the recent advances in the formulation techniques of liposomes, the asymmetric liposomes will find their way to the clinic, they mimic the biological membranes, and hence they can enhance drug uptake to diseased cells and retain therapeutic agents in lung tissues and other tissues. 

## Figures and Tables

**Figure 1 pharmaceutics-15-00294-f001:**
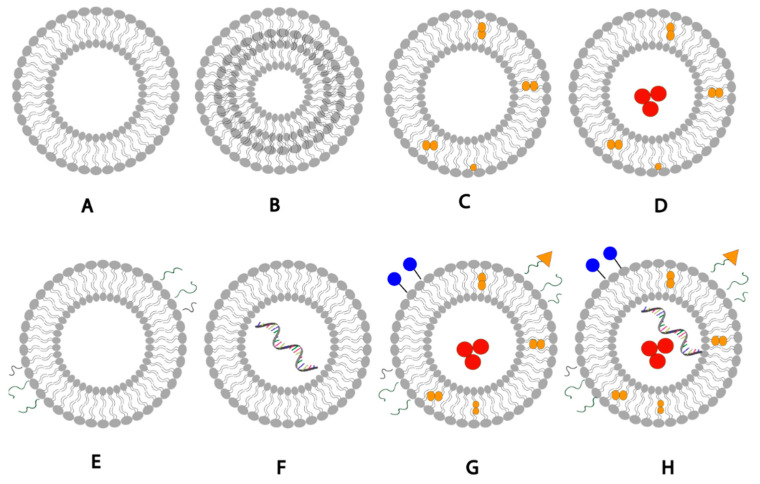
Schematic representation of liposomal drug delivery systems: (**A**) unilamellar liposome, (**B**) multilamellar liposome, (**C**) liposomes loaded with hydrophobic drug, (**D**) liposome loaded with hydrophobic drug in the bilayer membrane and hydrophilic drug in the aqueous core, (**E**) PEGylated liposomes with surface PEG polymer chains, (**F**) liposome loaded with mRNA, (**G**) liposome with surface conjugated drug, targeting ligands and PEG, hydrophilic and hydrophobic drugs, (**H**) liposome with surface conjugated drug, targeting ligands, PEG polymer chains, hydrophilic drug, hydrophobic drugs, and mRNA loaded. [8] [Reuse permitted by MDPI].

**Figure 2 pharmaceutics-15-00294-f002:**
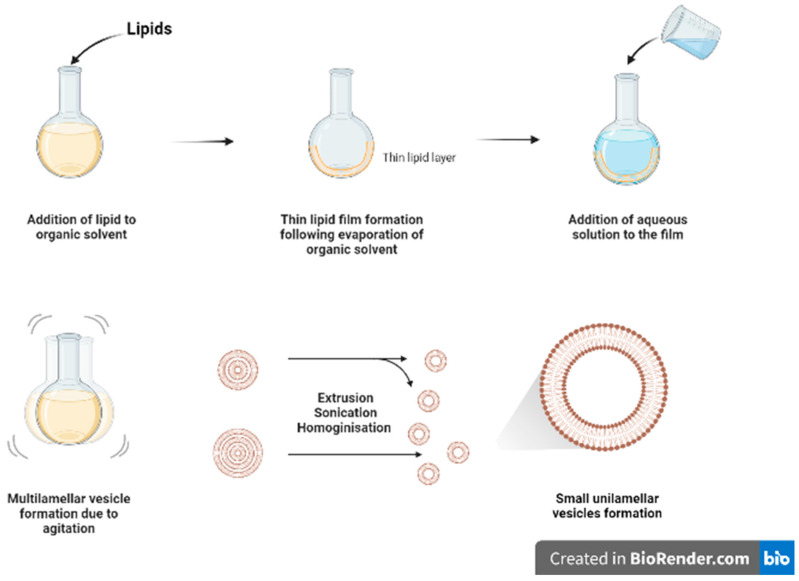
Schematic diagram of thin film hydration method.

**Figure 3 pharmaceutics-15-00294-f003:**
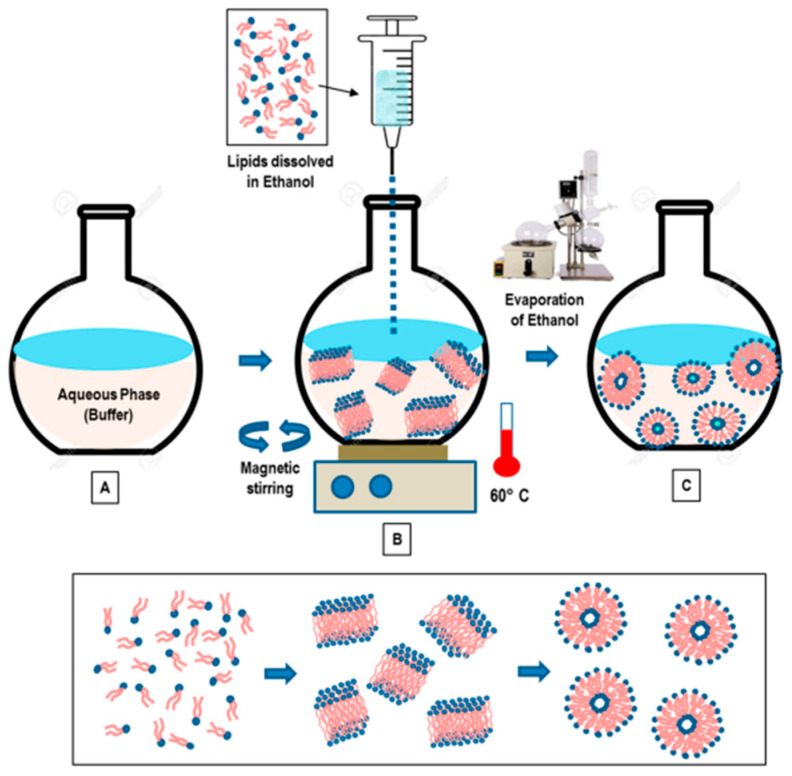
“Schematic representation of the main stages of the ethanol injection method” [9] [Reuse permitted by MDPI].

**Figure 4 pharmaceutics-15-00294-f004:**
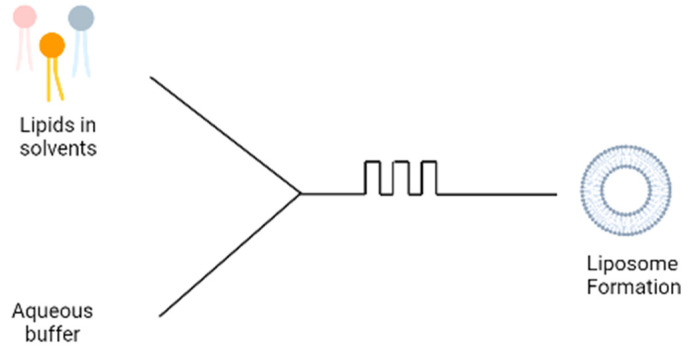
Schematic diagram of microfluidic technique.

**Figure 5 pharmaceutics-15-00294-f005:**
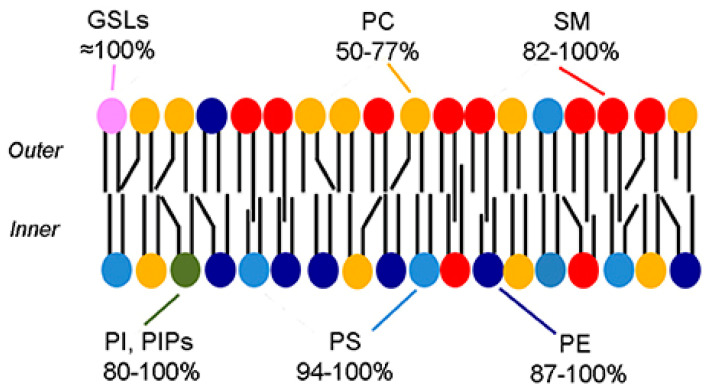
“Phospholipid asymmetry in the erythrocyte membrane” [22] [Reuse permitted by Creative Commons Attribution License (CC BY)].

**Figure 6 pharmaceutics-15-00294-f006:**
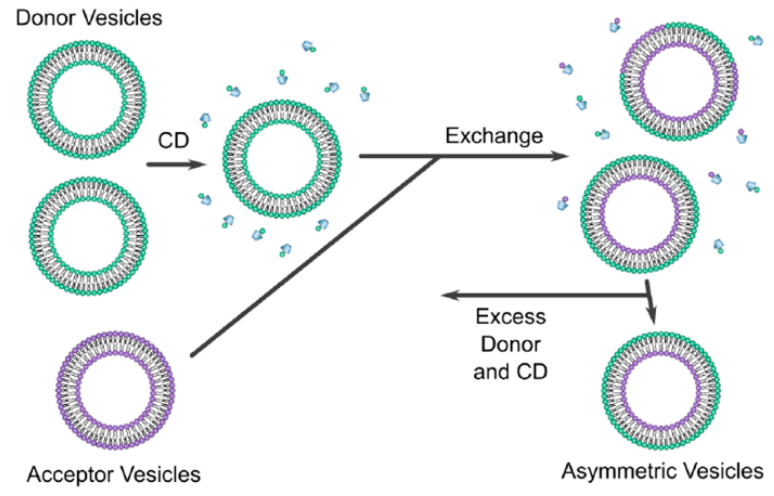
“Schematic of the CD protocol used to prepare asymmetric vesicles” [59] [Reuse permitted by Creative Commons Attribution License (CC BY)].

**Figure 7 pharmaceutics-15-00294-f007:**
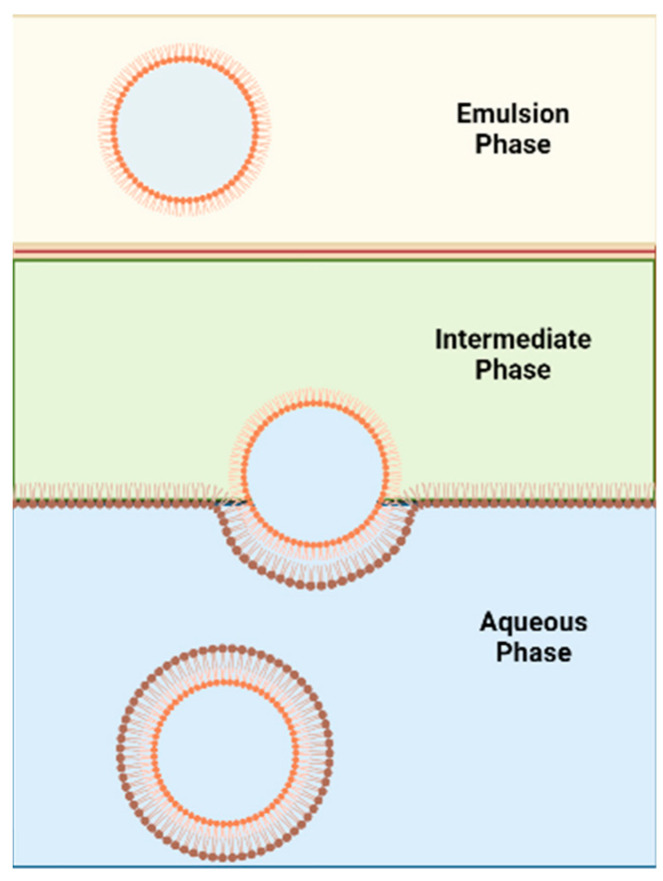
Schematic diagram of the inverse emulsion method.

**Figure 8 pharmaceutics-15-00294-f008:**
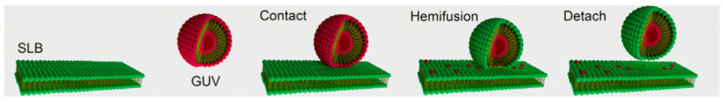
“Hemifusion yields asymmetric GUVs (aGUVs)” [20] [Reuse permitted by Elsevier].

**Table 1 pharmaceutics-15-00294-t001:** Advantages and disadvantages of symmetric liposomes formulation techniques.

Formulation Techniques	Advantages	Disadvantages
Thin film hydration	Simple and straightforward process Used for different kinds of lipid mixtures	Difficult to scale up Low entrapment efficiency for water soluble drugs Forms large vesicles with large size range Time-consuming
Ethanol injection	Reproducible, rapid and simple to use.	Difficult to remove all ethanol as it forms azeotrope with water.
Ether injection	Results in a concentrated liposomal suspension with improved entrapment efficiency	Inadequate mixing can result in heterogeneous liposomes Potential nozzle blockage
Reverse Phase Evaporation	Simple process Good encapsulation efficacy Allows the encapsulation of small, large and macromolecules	Requires large amount of organic solvent Not suitable for fragile molecules like peptides Time-consuming
Detergent removal	Good control of particle size Simple process Homogenous product	Produces low liposomal concentration Lipophilic drugs have Low entrapment efficiency Time consuming
Microfluidic	Simple process Allows particle size control	Difficult to remove the organic solvent Produces small amount of product

**Table 2 pharmaceutics-15-00294-t002:** Comparison between symmetric and asymmetric liposomes.

Item	Symmetric Liposomes	Asymmetric Liposomes
Compositions	Wide range of compositions and ratios	Need specific ratios and compositions Extra reagent or more than one preformed symmetric liposomes as template
Production methods and scalability	Established for production of different sizes and lamellarity Entrapment of small molecules and macromolecules (peptides and genes) For small scale (thin film hydration) and large scale (microfluidic technique) Good reproducibility in terms of characteristics and yields	Established for production of large unilamellar vesicles Nano-dimensions is emerging (e.g., pulse-jet flow) Successful entrapment with small molecules; potential for entrapment of large molecules with less chance of drug leakage and greater protection to labile drug More complex and with extra steps or reagents (for example, to enable lipid exchange) Custom made equipment/devices; scale up opportunity remains to be established
Characteristics, routes, and Stability	Prone to oxidation and hydrolysis related to lipid in use Good long term storage data Parenteral route is the main but can be adopted for all other routes Vesicles with targeting ability (versatile surface modifications)	Prone to lipid instability; potential interleaflet conversion Limited long-term stability data Targetable with different routes and better resembles to biological membranes
Physicochemical properties	size, shape, lamellarity, zeta potential and others.	Prove of asymmetry apart from standard tests.

**Table 3 pharmaceutics-15-00294-t003:** Advantages and disadvantages of asymmetric liposomes formulation techniques.

Formulation Techniques	Advantages	Disadvantages
Cyclodextrin exchange	Various types of phospholipid combinations can be used High encapsulation efficiency Components: lipids, cyclodextrins, sucrose	Considered violent when constructing vesicles. Can damage liposomes. Large amount of lipid content is lost. The loading of sucrose within the acceptor vesicle can lead to membrane tension and structural perturbations Requires removal of cyclodextrin and donor vesicle which could affect percentage yield of liposomes left in the formulation.
Reverse phase evaporation	High encapsulation efficiency Good scale-up ability Does not require exposing the formulation to high temperature	Requires dialysis which can be time consuming Requires ethanol which can inactivate many biologically active macromolecules and hinder their loading to liposomes
Ca^2+^ induced asymmetry	Has long stability of asymmetry which can last several days Does not require forming two different vesicles to produce asymmetry. Only one form of vesicles is required Simple techniques are used	Only negatively charged phospholipids can be used Long incubation time required (around 40 h)
The use of enzymes	Has long stability of asymmetry which can last four days Does not require forming two different vesicles to produce asymmetry. Only one form of vesicles is required Minimally invasive Can choose the lipids to modify without affecting other lipids within the outer leaflet of the vesicle Components: phosphatidylserine decarboxylase, lipids	The asymmetry formed is opposite of the biological membranes (PE is formed in the outer leaflet and not in the inner one) Can only work on specific phospholipids Enzymes can be denatured if exposed to wrong pH or temperature Requires the removal of the enzyme after achieving asymmetry which can be time consuming
Inverted emulsion technique	High degree of asymmetry Various types of phospholipid combinations can be used Simple techniques are used Good scale-up ability Components: lipids, glucose, oil, organic solvent	Presence of organic solvent between the lipid leaflets which can affect membrane’s physical and mechanical characteristics Low liposome formation yield Size of vesicles varies widely (polydispersity index can reach more than 20%)
Microfluidics	Has long stability of asymmetry for at least 6 weeks when using continuous microfluidic technique Automated method which reduces error and accelerate production of liposomes Components: lipids, organic solvent	Presence of organic solvent between the lipid leaflets which can affect membrane’s physical and mechanical characteristics Requires fabrication of a microfluidic device which can be time consuming and challenging
Hemifusion	Yields near 100% asymmetry without cyclodextrin or organic solvent contamination Yields vesicles with preserved vesicle content, with little leakage Components: fusogenic agent, lipids	It requires formation and observation of the SLB which can be time consuming Small amount of SLB lipid entered the inner layer of the liposome
Pulsed-jet flow	Has long vesicle stability for at least 7 days Able to produce cell-sized and nano-sized vesicles by increasing pressure and application time of pulsed-jet flows Components: lipids, organic solvent	Presence of residual organic solvent between the lipid leaflets which can affect membrane’s physical and mechanical characteristics Lipid content is lost as vesicles with large amount of organic solvent are discarded Requires fabrication of the pulsed microfluidic jet flow device which can be time consuming and challenging

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
