# Peer review of "Insights into Asymmetric Liposomes as a Potential Intervention for Drug Delivery Including Pulmonary Nanotherapeutics"

_pharmaceutics, 2023, doi:10.3390/pharmaceutics15010294_

Round 1

Reviewer 1 Report

The manuscript titled Insights into asymmetric liposomes as a potential intervention for drug delivery including pulmonary nanotherapeutics, in its current form, requires minor revisions to improve the rigor of this manuscript

-The manuscript is well written and organized

-Elaboration is needed throughout section 3. Kindly add more information about the articles being cited and the techniques that are used to fabricate liposomes

-Kindly add more information about pulsed-jet flow and microfluidic technique to make asymmetric liposomes

-Have you looked at asymmetric liposomes that are currently under clinical study? If yes, kindly add a table that summarizes all asymmetric liposomes that have been clinically tested

-Kindly add a table that compares symmetric and asymmetric liposomes in terms of their fabrication techniques, scale up abilities, advantages and disadvantages, etc

-Kindly include a section for challenges associated with asymmetric liposomes in terms of fabrication, stability, scale up, etc.

Author Response

The manuscript is well written and organized

Thank you

Elaboration is needed throughout section 3. Kindly add more information about the articles being cited and the techniques that are used to fabricate liposomes

More information on fabrication of liposomes with different techniques have been added and a table to compare methods has been included. Also, this section has been organised in a more readable manner, please refer to section 3.

Kindly add more information about pulsed-jet flow and microfluidic technique to make asymmetric liposomes

More information has been added, please refer to sections 7.2.2 and 7.2.4.

Have you looked at asymmetric liposomes that are currently under clinical study? If yes, kindly add a table that summarizes all asymmetric liposomes that have been clinically tested

Yes, we have searched the asymmetric liposomes and clinical trials, but those liposomes are not yet under clinical studies. Hope, this review will help scientists to find a way for those advantageous liposomes to be more formulated and investigated to be under clinical trials in the near future.

Kindly add a table that compares symmetric and asymmetric liposomes in terms of their fabrication techniques, scale up abilities, advantages and disadvantages, etc

Tables 1 and 2 have been added to cover symmetric and asymmetric liposomes, respectively, their advantages and disadvantages with different fabrication techniques. Section 8 was added to the manuscript for challenges associated with formulating asymmetric liposomes.

Kindly include a section for challenges associated with asymmetric liposomes in terms of fabrication, stability, scale up, etc

Section 8 was added to the manuscript for challenges associated with formulating asymmetric liposomes. Table 3 was added to reflect the advantages and disadvantages of asymmetric liposomes’ formulation techniques.

All changes are in red font colour.

Reviewer 2 Report

This review does not contribute much to the understanding of asymmetric vesicles. It reads as if it is an intro manual for working with lipids (not a bad thing, just not appropriate for the given issue). The review seems to lean heavily on the reference list from a previously published review on asymmetry and does not cover the substantial advancements since 2016. In the context of drugs, the approach by London is not a viable approach when compared to what Heberle et al. (2016). There is a plethora of recent literature here that is being overlooked.

The dispersion methods are a distraction and do not add anything. If there was discussion on how lipids impact dispersion that could be useful. There is a recent paper in Biophysical Journal that looks at the importance of charged lipid when forming ULVs by Scott et al..

There does not seem to be an in depth discussion of the current landscape of  synthetic asymmetric vesicles, especially those properly characterized and stress-free.

There has been substantial work on the stability of asymmetric vesicles on their own and in the presence of perturbants. There are numerous neutron scattering studies dating back to the mid to late 2000s to present. There are also NMR studies of recent.

I recommend that the author go back to literature and read past Marquardt 2015. Granted there are some more recent articles, but nothing that has revolutionized the field.

Author Response

This review does not contribute much to the understanding of asymmetric vesicles. It reads as if it is an intro manual for working with lipids (not a bad thing, just not appropriate for the given issue). The review seems to lean heavily on the reference list from a previously published review on asymmetry and does not cover the substantial advancements since 2016. In the context of drugs, the approach by London is not a viable approach when compared to what Heberle et al. (2016). There is a plethora of recent literature here that is being overlooked.

The manuscript was proof-read. The manuscript has been reviewed extensively and relevant information has been added to include more recent literature and relevant description of methods to formulate asymmetric liposomes, All changes are made in red colour, for example refer to section 7.

The dispersion methods are a distraction and do not add anything. If there was discussion on how lipids impact dispersion that could be useful. There is a recent paper in Biophysical Journal that looks at the importance of charged lipid when forming ULVs by Scott et al..

The methods have been written and organised in more organised manner, the manuscript has been revised, please refer to red colour font in the manuscript.

There does not seem to be an in depth discussion of the current landscape of  synthetic asymmetric vesicles, especially those properly characterized and stress-free.

Section 7 has been revised to reflect on that and also Tables 1, 2 and 3 were added to the manuscript.

There has been substantial work on the stability of asymmetric vesicles on their own and in the presence of perturbants. There are numerous neutron scattering studies dating back to the mid to late 2000s to present. There are also NMR studies of recent.

Information about NMR and neutron scattering studies have been added, please refer to Section 9.

I recommend that the author go back to literature and read past Marquardt 2015. Granted there are some more recent articles, but nothing that has revolutionized the field.

Done, and some more recent articles have been added, the research is never ending, and this review may help researchers to critical think to innovate asymmetric liposomes formulation to add to the field.  

All changes are in red colour font.

Reviewer 3 Report

This review article focused on the asymmetric lipid vesicles as a potential clinical intervention drug delivery system. The authors described the method of fundamental liposome formation, the explanation of biological cell membranes, the formation methods of nano-sized asymmetric liposomes and the prospects of nano-sized asymmetric liposomes for the drug delivery system. Nevertheless, the following issues should be addressed before the manuscript can be accepted for publication.

1.       The various methodologies of nano-sized asymmetric liposome formations were described in section 7. The methodologies of asymmetric liposome formation in section 7.6 and 7.7 were the cell-sized liposomes. Was there the formation of nano-sized asymmetric liposomes in the methodologies of section 7.6 and 7.7? It would be very useful for readers if the author can summarize each formation methodology of nano-sized asymmetric liposomes in a table including the liposome size, composition of asymmetric lipid membranes, and encapsulation of reagents.

2.       Section 7.2 Pulsed-jet flow (Kamiya et al (2021)) was modified from the method of cell-sized asymmetric lipid vesicles (Kamiya et al., Cell-sized asymmetric lipid vesicles facilitate the investigation of asymmetric membranes” Nature Chemistry, Vol. 8 (2016) pp.881-889.). Please refer to Nature Chemistry.

Author Response

  1. The various methodologies of nano-sized asymmetric liposome formations were described in section 7. The methodologies of asymmetric liposome formation in section 7.6 and 7.7 were the cell-sized liposomes. Was there the formation of nano-sized asymmetric liposomes in the methodologies of section 7.6 and 7.7? It would be very useful for readers if the author can summarize each formation methodology of nano-sized asymmetric liposomes in a table including the liposome size, composition of asymmetric lipid membranes, and encapsulation of reagents.

Section 7, This method section for asymmetric liposomes has been divided into two categories: formulating nano-sized and formulating cell-sized liposomes, and more information with new references have been added, the new references have been added to the reference list as well.

All changes have been made in red colour.

A table has been added to include the advantages and disadvantages of the techniques for asymmetric liposomes formulation (Table 3).

  1. Section 7.2 Pulsed-jet flow (Kamiya et al (2021)) was modified from the method of cell-sized asymmetric lipid vesicles (Kamiya et al., “Cell-sized asymmetric lipid vesicles facilitate the investigation of asymmetric membranes” Nature Chemistry, Vol. 8 (2016) pp.881-889.). Please refer to Nature Chemistry.

This has been reviewed and more information with new references have been added, the new references have been added to the reference list as well.

All changes have been made in red colour.

Round 2

Reviewer 2 Report

My concerns have been addressed.